# Validity Evidence of the Multidimensional Emotional Disorders Inventory among Non-Clinical Spanish University Students

**DOI:** 10.3390/ijerph18168251

**Published:** 2021-08-04

**Authors:** Jorge Osma, Víctor Martínez-Loredo, Alba Quilez-Orden, Óscar Peris-Baquero, Carlos Suso-Ribera

**Affiliations:** 1Departamento de Psicología y Sociología, Universidad de Zaragoza, 44003 Teruel, Spain; loredo@unizar.es (V.M.-L.); 699298@unizar.es (A.Q.-O.); operis@unizar.es (Ó.P.-B.); 2Instituto de Investigación Sanitaria de Aragón, 50009 Zaragoza, Spain; 3Unidad de Salud Mental Moncayo, 50500 Tarazona, Spain; 4Departamento de Psicología Básica, Clínica y Psicobiología, Universitat Jaume I, 12071 Castellón de la Plana, Spain; susor@uji.es

**Keywords:** dimensional psychopathology, transdiagnostic, emotional disorders, assessment, sources of validity evidence, university students

## Abstract

The current diagnostic systems for mental health disorders are categorical, which, it has been argued, poorly reflect the reality of mental health problems. This is especially relevant in emotional disorders (EDs), especially due to the existing comorbidity between supposedly different disorders. To address this, Brown and Barlow developed a hybrid dimensional−categorical approach to EDs that can be evaluated with the Multidimensional Emotional Disorder Inventory (MEDI), a transdiagnostic self-report questionnaire. This study aims to adapt and explore the sources of validity evidence of the MEDI in a non-clinical sample of Spanish university students (*n* = 455). Two confirmatory analyses were performed: one with a four-dimensional structure obtained with an exploratory analysis and another with the original nine-dimensional structure of the MEDI. The latter obtained a better fit. The descriptive data, including percentiles, T-scores, and sex differences in total scores are also provided, together with sources of validity evidence. These revealed significant moderate interrelations between factors and with related measures (e.g., personality, depression, and anxiety). This study adapted the MEDI for use in Spanish, provides further support about its factor structure, and offers novel data about its validity sources. The MEDI makes the evaluation of dimensional and transdiagnostic models easier, which might be fundamental in present and future research and clinical practice.

## 1. Introduction

The current diagnostic systems for mental health disorders, mainly represented by the Diagnostic and Statistical Manual of Mental Disorders (DSM-5) [1] and the International Classification of Disease (ICD-11) [2], are categorical. This means that that they divide psychopathology in as many diagnoses as are possibly establishable [3]. Both research and clinical practice have criticized this approach due to the disproportionate number of potentially artificial categories into which psychopathology is divided [4].

The problems associated with the categorical classification of psychological disorders are especially relevant for emotional disorders (hereinafter, ED), which include anxiety and depression disorders [5]. Several reasons justify the detrimental impact of categorical classification systems for EDs. First, there is a substantial overlap in the characteristics that define different diagnoses in the current categorical classification systems, such as marked fear and anxiety in social anxiety disorder and specific phobia [3]. There is only modest agreement among professionals when classifying phenotypes shared by multiple diagnostic categories [6]. The lifetime comorbidity of anxiety and mood disorders is as high as 75%, which supports the existence of communalities between them [3]. Finally, a large number of people receive a diagnosis of “not otherwise specified disorder” because they only meet a few of the necessary criteria for a given diagnosis [3]. However, many of these individuals do present clinically significant levels of impairment even if they fail to meet all the requirements for a proper diagnosis [3].

All these limitations of the categorical classification system of mental health problems are, in fact, well documented and have been debated for years, especially in relation to EDs [6]. As a consequence, several authors have argued that this group of disorders should be conceptualized as dimensional, as opposed to discrete constructs, which have more similarities than differences [7].

In order to overcome these limitations of categorical classifications, several dimensional approaches have been developed in the past decades. Two of the most popular are the Research Domain Criteria (RDoC) [8] and the Hierarchical Taxonomy of Psychopathology (HiTOP) [9]. The RDoC is a transdiagnostic approach that aims to study and explore the functioning aspects associated with mental health problems. As such, it includes dimensions of psychological processes that underlie a continuum between normal to abnormal functioning [8]. Importantly, functioning according to this approach is primarily defined by means of biological processes. Therefore, it is difficult to imagine how the RDoC could become a classification system to be currently adopted in routine clinical care [6]. Different to the RDoC, the HiTOP is a quantitative and dimensional classification system based on a multilevel organization [9]. In general, the dimensions of the HiTOP are more descriptive and specific than those in the RDoC. However, the existence of these specific dimensions makes its clinical usefulness unclear and would require using several specific questionnaires to evaluate each dimension [6]. These limitations of dimensional approaches have become especially relevant at the clinical practice level [7]. As a result, categorical diagnostic categories have remained to be the mainstream way to facilitate communication between clinicians [6].

Considering the empirical limitations of purely categorical models and the practical limitations of dimensional approximations, Brown and Barlow [10] developed a hybrid dimensional−categorical approach to the assessment of EDs. This was done in an attempt to offer a mixed vision that combined the fundamental advantages of both approaches. Specifically, Brown and Barlow chose to differentiate a series of factors shared by EDs. They argued that, based on the scores of such factors, they would be able to reveal a unique profile for each patient, which may be closer or more distant to the diagnostic categories provided by traditional classification systems. These factors are neuroticism, anxiety or behavioral inhibition (NT), behavioral activation or positive affect (PT), depressed mood (DM), autonomic arousal (AA), somatic anxiety (SOM), social anxiety (SEC), intrusive cognitions (IC), traumatic re-experiencing and dissociation (TRM), and avoidance (AVD) [10].

The use of these constructs has been argued to make the evaluation of the patient’s clinical profile more efficient. For example, the use of such factors facilitates the identification of mechanisms underlying the symptoms that cause clinically significant interference. These constructs also facilitate to obtain information about the severity of the patient in terms of dispositional and more proximal factors that influence behavior and emotional status. Finally, the use of the aforementioned higher and lower order factors appears to increase the reliability and validity of the established diagnosis because it facilitates differential diagnosis and reduces comorbidity rates by focusing on dimensional aspects that characterize groups of individuals that share a common diagnosis [7]. Probably due to the novelty of this approach and the dominance of categorical diagnostic approaches, there still little information on the validity of the mixed dimensional−categorical approaches to psychopathology [6]. Another explanation, however, lies in the need to include several self-report measures to assess every dimension included in the profile. This would require a large amount of time and associated costs, as well as a significant burden for clinicians/researchers and patients/participants. For this reason, authors have argued that it is essential to develop standardized assessment tools to be used for the study of dimensional classifications [6].

The Multidimensional Emotional Disorder Inventory (hereinafter, MEDI) was created with the previous goal in mind. The MEDI is a self-report questionnaire that aims to evaluate the 9 transdiagnostic dimensions proposed in the hybrid dimensional−categorical approach for the classification of ED mentioned above. Importantly, it does so in an efficient manner, that is, by providing a brief evaluation with the fewest number of items possible [6]. The MEDI allows us to obtain an overview of the functioning of the patient across nine dimensions (NT, PT, DM, AA, SOM, SEC, IC, TRM, and AVD). Importantly, this information about different constructs shared by EDs is obtained in a single instrument, which eliminates the need to administer different, specific self-reports for each construct [3]. Thus, the MEDI is particularly useful in cases where there is high comorbidity, such as in EDs [3].

Because the lower and higher order dimensions evaluated by the MEDI are well established and based on both research and clinical practice, this self-report has several advantages: (1) it promotes research on the dimensional−categorical approach to the classification of EDs, which allows empirical data to be obtained on the effectiveness and efficiency of the categorical classification systems; (2) it makes it possible to study the severity and interference of symptoms based on a shared dimension or multiple dimensions, as opposed to focusing on different specific symptoms of each disorder; (3) it allows clinicians to obtain specific information that can be expanded using other techniques (functional analysis); (4) it facilitates treatment planning (an underlying mechanism that can be used as treatment targets), which favors the prioritization of therapeutic objectives; (5) finally, it allows a follow-up evaluation of the changes obtained in the different dimensions during treatment, without the need to administer a larger set of questionnaires [6].

So far, only one study has evaluated sources of validity evidence of the MEDI [6]. Thus, more research in this regard is needed, both in clinical and in community samples. In this sense, the aforementioned study by Rosellini and Barlow [6] evaluated the characteristics of the MEDI in a clinical sample, so their conclusions are more relevant to the field of diagnosis and the establishment of a therapeutic intervention plan [6].

Obtaining data on the validity of the MEDI in a community sample would allow one to focus on prevention and early detection, as opposed to treatment. In particular, it could help to detect a population vulnerable to developing EDs, establish risk profiles, and develop personalized prevention programs.

EDs are the most prevalent disorders worldwide [11], but are especially prevalent in young people [12]. Specifically, the university period is experienced as a particularly stressful life stage, in which it is necessary to adapt to numerous potentially stressful situations, such as changes in housing, city of habitation, meeting new people, and studying more complex matters. Ultimately, this may increase the symptoms of psychological distress and the incidence of mental disorders in this population [13]. Some studies show that ED rates are higher among university students than in other populations, such as non-university young adults or older adults [14,15,16]. Therefore, the university context could have a strong impact on the psychological well-being of individuals, which makes this an ideal context for the early detection of vulnerable profiles and the development of prevention programs [17].

In this sense, the development and validation of dimensional evaluation instruments for EDs in this population group is an essential task if we aim to detect, evaluate, and prevent the development of ED among young adults [18]. The validation of a dimensional instrument such as the MEDI would also allow future investigation using dimensional approaches in this population. Specifically, adapting and validating the MEDI would facilitate preventive interventions based on hybrid approaches and would permit easy monitoring of such interventions, thanks to its self-report format.

The present study adapts the MEDI into the Spanish language and investigates its psychometric characteristics in a non-clinical sample of Spanish university students. The main objectives of the study are: (1) to adapt the MEDI into Spanish using a back-translation process; (2) to examine the internal structure of the questionnaire; (2) to estimate the reliability of each subscale in terms of their internal consistency; (3) to examine its sources of validity evidence in relation to other variables; (4) to explore sex differences in total scores; (5) and to report potentially normative data in percentiles and T-scores. Overall, we expect to replicate the original nine-factor structure of the MEDI and to find evidence of the internal consistency and validity of the MEDI subscales in relation to other measures. Sex differences will be explored in an exploratory manner and discussed according to previous literature with similar constructs.

## 2. Materials and Methods

### 2.1. Participants

The sample comprised 507 university students who agreed to participate in the study and completed the battery of questionnaires. Of these, 52 participants were excluded from the analysis because they were receiving psychological/psychiatric treatment at the time of the evaluation. The final sample was composed of *n* = 455 participants, with a mean age of 22.92 years (*SD* = 5.95, range 18–57). Regarding sex, 85.1% of them were females (*n* = 387). The sociodemographic information is reported in Table 1.

### 2.2. Instruments

Sociodemographic data. Information was collected on age, sex, marital status, educational level, employment status, ongoing university studies, and both current and history of psychological/psychiatric treatment.

The Multidimensional Emotional Disorder Inventory (MEDI; [6]). The MEDI consists of 49 items with a Likert-type response format ranging from 0 (not at all characteristic of me) to 8 (totally characteristic of me). It evaluates 9 transdiagnostic dimensions: (1) Neurotic Temperament (e.g., Item 1: “I get upset by trivial things”); (2) Positive Temperament (e.g., Item 2: “It doesn’t take much to make me laugh”); (3) Depressive Mood (e.g., Item 11: “I feel sad and blue”); (4) Autonomic Arousal (e.g., Item 4: “I have been experiencing breathlessness”); (5) Somatic Anxiety (e.g., Item 19: “I worry about my health”); (6) Social Anxiety (e.g., Item 7: “I am uncomfortable mingling at social events”); (7) Intrusive Thoughts (e.g., Item 5: “Other people would consider some of my thoughts to be odd”); (8) Traumatic Re-experiencing (e.g., Item 8: “I cannot stop thinking about horrific things that I have experienced of seen”); and (9) Avoidance (e.g., Item 9: “I cope with unpleasant thoughts, feelings, or images by trying to distract myself”). Table 2 shows the criterion/definition used to create each scale both in the original development study [6] and in the present investigation.

The NEO Five-Factor Inventory (NEO-FFI; [19]). The NEO-FFI is a self-administered questionnaire that includes 60 items. Each item has a 5-point Likert-type response scale ranging from 0 (totally disagree) to 4 (totally agree). The NEO-FFI evaluates the dimensions of personality included in the Big Five Factor Model, that is, Neuroticism, Extraversion, Openness to experience, Conscientiousness, and Agreeableness. For this study, only the 24 items of the Neuroticism and Extraversion subscales were used (12 items for each dimension). The internal consistency estimates of Neuroticism and Extraversion in the present study were α = 0.88 and α = 0.88, respectively.

The Anxiety Sensitivity Index-3 (ASI-3; [20,21]). The ASI-3 is an 18-item self-administered questionnaire that evaluates three components of anxiety: Physical, Cognitive, and Social anxiety. Responses use a 5-point Likert scale ranging from 0 (Not at all applicable to me) to 4 (Very much). In the present study, the 6 items related to the physical anxiety subscale were used. The Cronbach’s alpha of the physical anxiety dimension in the present study was *α* = 0.86.

The brief version of the Fear of Negative Evaluation Scale (BFNE; [22,23]). The BFNE consists of 12 items that use a 5-point Likert-type response format ranging from 1 (Not at all characteristic of me) to 5 (Extremely characteristic of me). For this study, the 8 items of the straightforward scale were used, as proposed by Gallego-Pitarch [23]. The Cronbach’s alpha of the straightforward scale in the present sample was *α* = 0.95.

The Depression, Anxiety and Stress Scale (DASS-21; [24,25]). The DASS-21 is a 21-item self-administered questionnaire that evaluates Depression, Anxiety, and Stress. Items use a 4-point Likert-type response format ranging from 0 (Did not apply to me at all) to 3 (Applied to me very much, or applicable most of the time). For this study, only the Depression and Anxiety subscales were used (7 items for each dimension). These subscales will be referred to as DASS-14 throughout the manuscript. The Cronbach’s alpha estimates in the present study were α = 0.88 for Depression and α = 0.79 for Anxiety.

The Compulsive Inventory-Revised (OCI-R; [26,27]). The OCI-R assesses the Obsessive−Compulsive Disorder dimensions of Cleaning/Washing, Checking, Order, Obsessions, Hoarding, and Neutralization. It consists of 18 items grouped into a single total score. Items use a 5-point Likert-type response format ranging from 0 (Not at all/None/Not at all) to 4 (Very much). The internal consistency of the total score in the present study was α = 0.87.

The Davidson Trauma Scale (DTS; [28,29]). The DTS is an 18-item questionnaire that assesses the frequency and severity of post-traumatic stress disorder symptoms. The response format is a 5-point Likert-type ranging from 0 (Never/Not at all) to 4 (Daily/Extreme). The Cronbach’s alpha in the present sample was α = 0.95 for the frequency dimension and α = 0.95 for the severity dimension.

The Brief Experiential Avoidance Questionnaire (BEAQ, [30,31]). The BEAQ is a 15-item questionnaire that assesses Experiential Avoidance. Responses use a 6-point Likert scale ranging from 1 (Strongly Disagree) to 6 (Strongly Agree). The Cronbach’s alpha of the BEAQ in the present sample was α = 0.88.

### 2.3. Procedure

The participants in this study were recruited at different universities in Spain where they were conducting their university studies. Inclusion criteria were: (1) being aged 18 or over; (2) conducting their university studies in Spain; (3) being fluent in Spanish, and (4) signing the informed consent form. The only exclusion criteria were being under psychological or psychiatric treatment at the moment of the assessment.

Participants were recruited by professors working at different universities in Spain who agreed to collaborate in the study. In this case, professors sent advertisement messages to their students via email or using the intranet campus. Additionally, university students were recruited using social networks and advertisements at the university campus. In all cases, a link to the online survey platform Qualtrics was available for the students to access the study [32]. Student eligibility was screened in the first page of the online survey. The informed consent was also provided online. Once eligibility was screened and participants consented to participate, they were asked to complete a set of questionnaires (see the Instruments section), including the MEDI. The time to complete the questionnaires took approximately 20 to 30 min. Participation in the study was voluntary, anonymous, and did not include any financial compensation. The research was approved by the Research and Ethics Committee of the (blinded) University.

The Spanish version of the MEDI was obtained after translation into Spanish and a back translation into English to ensure conceptual equivalence. Following the International Test Commission recommendations [33], the translation into Spanish was carried out by two independent researchers who were proficient in English. These researchers compared and corrected discrepancies in the translations. This Spanish version was back translated into English by two independent English native translators who were fluent in Spanish. No significant differences were found between the original and the obtained version and no changes were required for the final Spanish version. Appendix B includes the Spanish adaptation of the MEDI.

### 2.4. Data Analysis

The sample was randomly divided into two subgroups. After performing an exploratory factor analysis (EFA) in the first one (*n* = 228), the proposed structure was cross validated via a confirmatory factor analysis (CFA) of the second one (*n* = 227). Considering the items’ response scale and the original factor structure of the MEDI [6], the EFA was performed using the matrix of Pearson correlations, the unweighted least squares estimator (ULS) extraction method, and a Promin rotation [34]. The optimum number of factors was determined by the optimal implementation of parallel analysis [35], based on 1000 random resampling operations. Goodness of fit was examined with the goodness of fit index (GFI < 0.95; [36]) and the root mean square of the residuals (RMSR < 0.08; [37]). The estimation method for the CFA was the maximum likelihood method (MLM). Goodness of fit was examined with the comparative fit index (CFI > 0.90; [37]), the RMSR, the root mean square error of approximation (RMSEA < 0.08), and the χ^2^/df (<2). Because the parallel analysis suggested an internal structure that differed from the original one, two CFAs were conducted. In the first CFA, we set the structure suggested by the EFA and, in the second one, the original one proposed by Rosellini and Brown [6]. Both solutions were compared using the following indices: CFI, χ2/df, Aikaike information criteria (AIC), and sample-adjusted Bayesian information criteria (SABIC). When needed to improve the model fit, modification indices were used to select items where the correlation between errors was recommended. The modification indices estimate the magnitude of reduction in the chi-square statistic. Adding the changes with the highest modification value into the model to be freely estimated is a data-driven post hoc approach widely use in the literature.

The reliability of the factors was estimated using the Cronbach’s alpha estimate. To obtain evidence of validity in relation to other variables, Pearson zero-order correlations were performed between the MEDI factors and Neuroticism, Extraversion, the ASI, the BFNE, the DASS-14 subscales, the OCI-R, the DTS, and the BEAQ. Sex differences in total scores were explored via a multivariate analysis of variance. Percentiles and T-scores [38] scaling are also provided.

Data analyses were performed using the statistical software packages SPSS v26 (Chicago, IL, USA), FACTOR 10.10.03 (Tarragona, Spain), and Mplus 8 (Los Angeles, CA, USA).

## 3. Results

### 3.1. Validity Evidence Based on the MEDI Internal Structure: EFA and CFA

The Kaiser−Meyer−Olkin test (index = 0.911), the Bartlett Sphericity test (*χ*^2^(1176) = 2415.8, *p* < 10^−5^) suggested the adequacy of the exploratory analysis and the model fit. A four-factor solution according to the parallel analysis appeared as the best solution, with the percentage of explained variance per factor ranging 4.78–33.27% and adequate goodness of fit indices (GFI = 0.993; RMSR = 0.030) (see Appendix A for factor loadings).

Two CFAs were performed in the second sample: the first tested the structure proposed by the EFA (four-factor solution) and the second tested the original nine-factor structure proposed by Rosellini and Brown [3]. The results suggested a better fit of the original structure (χ^2^/df = 1.59, CFI = 0.865; RMSR = 0.074; RMSEA = 0.051) over the one proposed by the EFA (χ^2^/df = 2.01, CFI = 0.764; RMSR = 0.09; RMSEA = 0.067). In addition, the AIC and the SABIC were lower in the original structure (AIC: 43,407.59 vs. 43,971.15; SABIC: 43,454.37 vs. 44,010.26), suggesting its parsimony. Even though the original nine-factor solution was a better model, its CFI was slightly below the commonly accepted minimum threshold. Thus, and considering that the AFE and the CFA solutions were tested in independent samples, a CFA using the original structure was performed on the whole sample (*n* = 455). This was done to test the effect of sample size on model fit. The results showed an improvement in the model fit when including the whole sample (CFI = 0.89; ΔCFI = 0.025) and when correlating the errors of items 19 and 38 and items 49 and factor 9 (as suggested by the modification indices; CFI = 0.90; ΔCFI = 0.035). Considering this, the original structure was retained. Nonetheless, replication of the following analyses using the four-factor solution obtained with the EFA is reported as a Appendix A.

The MEDI subscales showed excellent reliability indices as estimated with the internal consistency of the nine factors (Cronbach’s alphas between 0.74 and 92). Item factor loadings and discrimination indices are shown in Table 3 (See Appendix A for reliability indicators based on the four-factor structure suggested by the EFA).

### 3.2. Validity Evidence Based on Relationships with Other Variables

All questionnaires were significantly and positively correlated excepting for the PT and Extraversion, which were negatively correlated with all the other scales. The largest associations occurred between DM and the DASS-14-Depression scale (*r* = 0.81, *p* < 0.001), NT and Neuroticism (*r* = 0.73, *p* < 0.001), SOC and Extraversion (*r* = −0.72, *p* < 0.001), and between IC and Neuroticism (*r* = 0.70, *p* < 0.001). In addition, we observed significant correlations between the MEDI subscales and the corresponding/expected self-report measures (Appendix A), namely NT and Neuroticism (*r* = 0.73); PT and Extraversion (*r* = 0.59, *p* < 0.001); DM and DASS-14-Anxiety (*r* = 0.81, *p* < 0.001); AA and DASS-14-Anxiety (*r* = 0.74, *p* < 0.001); SOM and ASI (*r* = 0.53, *p* < 0.001); TRM and DTS (*r* = 63, *p* < 0.001); and AVD and BEAQ (*r* = 0.67, *p* < 0.001). The only exceptions to the previous were the SOC, which was more significantly associated with Extraversion (*r* = −0. 72) than with BFNE (*r* = 0.49, *p* < 0.001), and the IC, which correlates more strongly with Neuroticism (*r* = 0.70), the DASS-14-Depression (*r* = 0.58, *p* < 0.001), the BEAQ (*r* = 0.53, *p* < 0.001), and the DTS (*r* = 0.53, *p* < 0.001) than with the OCI-R (*r* = 0.52, *p* < 0.001).

In relation to the four-factor structure, the largest associations occurred between F1 and F4 (*r* = 0.75, *p* < 0.001), F1 and Neuroticism (*r* = 0.72, *p* < 0.001), F2 and Extraversion (*r* = −0.68, *p* < 0.001), F4 and Neuroticism (*r* = 0.73, *p* < 0.001), and F4 and DASS-14-Anxiety (*r* = 0.66, *p* < 0.001). Pearson zero-order correlations between the MEDI subscales, Neuroticism and Extraversion, ASI, BFNE, DASS-14 subscales, OCI-R, DTS and BEAQ are shown in the Supplementary Material for both the original (Appendix A) and four-factor (Appendix A) structures.

### 3.3. Sex Differences and Scaling of the MEDI

Table 4 shows the distribution of the MEDI subscales across the whole sample and divided by sex (see Appendix A for the same information with the four-factor solution). Females scored significantly higher both in neurotic (M = 18.33) and positive temperament (M = 28.80) compared to males (M = 15.86 and 27.03, respectively; *d* = 0.27 in both cases). Males scored significantly higher in depressed mood (M = 10.98 vs. 8.63; *d* = 0.27).

Normative data in percentile ranks and T-scores are shown in Table 5 (see Appendix A for normative data based on the four-factor solution). An example of the MEDI profile in two participants is shown in Figure 1 (See Appendix A for their profile based on the four-factor solution).

## 4. Discussion

This is the first study to adapt the MEDI into Spanish, to investigate its internal structure, and to explore sources of validity evidence in non-clinical samples. We hypothesized that we would replicate the original nine-factor structure proposed by Rosellini and Brown [6] and expected to obtain good internal reliability indices and evidence on the validity of the MEDI subscales in relation to other measures. Exploratory analysis suggested an internal structure composed by four factors. However, a confirmatory analysis with the original nine-factor structure indicated that this model had a better fit with adequate parsimony indices. Overall, this solution was found to be more desirable not only considering fit indices, but also to adhere to the theoretical model proposed in the development of the MEDI. In addition to the factor structure, which obtained good internal consistency indices, the results generally supported the validity of the MEDI subscales in relation to other measures. Specifically, most of the subscales correlated strongly with the corresponding measures that were selected for this purpose (see Table 2). In sum, these findings in university students support grouping the MEDI items using the original nine dimensions proposed by Rosellini and Brown [6]. Importantly, the validity results indicate that these dimensions are likely to measure the constructs they were designed to evaluate.

As noted in the previous lines and in the results section, the results of the exploratory factor analyses supported a four-factor solution to the MEDI, while a confirmatory analysis indicated a better fit of the original nine-factor solution. Several explanations may exist for the findings. For example, the sample of the present study was non-clinical, which means that scores in some subscales such as IC or TRM may be less well represented, since cognitive intrusions or traumatic re-experiencing are less frequent in the general population. Future studies in other non-clinical samples are important to provide further data about this hypothesis. Another reason that may explain the preference for a four-factor solution in the exploratory analysis lies in parsimony. In particular, it is possible that a four-factor solution obtained the better parsimony-fit ratio. While acknowledging this, it is important to note that the nine-factor solution resulted in adequate fit and parsimony evidence. In addition, this solution is more consistent with the theoretical model used to develop the MEDI, which justifies why this solution was ultimately preferred to the four-factor one.

As in the original study [6], the present study provides strong support for the internal consistency of the MEDI factors, with Cronbach’s alphas ranging between 0.74 and 0.92. The interrelationships between the different factors were between moderate and large, consistent with the results obtained in the original scale development study [6]. For example, the NT and PT subscales were inversely correlated, which is consistent with other studies that show inverse correlations between neuroticism and extraversion [6,39]. Following the first and second order classification [10], we can see how the NT correlates positively with all other lower order variables, while PT inversely correlates with them. Importantly, the correlations between the different MEDI subscales were strong. This is consistent with the hybrid categorical−dimensional model proposed for Brown and Barlow [10], which considers that the factors are shared by different categorical diagnoses that present high rates of comorbidity.

An important contribution of the present study was that the DSM-5 diagnostic criteria was used. This was already recommended by the authors who developed the MEDI [6]. In addition, we updated the sources of validity evidence accordingly (see Table 2), which should also be seen as a strength of the study. In this sense, the positive correlations of the MEDI subscales with the majority of the corresponding specific self-reports are an important finding of the present investigation that support the idea that the MEDI evaluates what it is expected to measure. Still in relation to the sources of validity evidence, we want to note that, thanks to the combination of the MEDI and the specific measures used to evaluate its validity evidence, we can see the utility of the MEDI in terms of item reduction. In particular, while the MEDI is composed of 49 items, we required 88 items to evaluate all the constructs included in the MEDI by means of traditional self-reports (see Table 2). This implies a reduction of 39 items and a single set of instructions and response scales, which favors a more efficient evaluation in terms of time and the associated costs and assessment burden derived from the evaluation of EDs. In addition, the MEDI introduces the advantages of dimensional assessment mentioned in the introduction [6,7].

Regarding sex differences, we observed that women presented higher levels of NT and PT. These findings are consistent with past research showing that scores in neuroticism and extraversion are higher in females [40]. Note, however that these differences in our study were small, so the interpretation should be made accordingly. Contrary to NT and PT, DM scores in our study were higher in males compared to females. Again, however, the differences were small in size. These findings are not consistent with past studies, in which scores in depressive symptoms have been higher in women [41]. This result could be due to the low number of men in the study sample, only 14.9%.

In the current study, we provided data on percentile ranges and T-scores. As a result of this, future studies will be able to determine the relative position of a given person when compared to our data, which of course is not representative of the general population in Spain because it was obtained from university students. This information, together with the possibility to construe a profile with the MEDI, as exemplified in this study, will allow researchers and clinicians to perform a dimensional evaluation of the characteristics of individuals in the future. As noted in the introduction, this becomes fundamental considering the disadvantages of the categorical classification systems. In particular, thanks to this evaluation system, clinicians and researchers will be able to monitor how the scores in different dimensional constructs change after applying a prevention or an intervention transdiagnostic program. Importantly, our normative scores may be useful when applying such programs in a university context in Spain, as the sample was composed of students from universities across the country.

The present study has a number of limitations. One of them refers to the sample size. Specifically, by dividing the sample into two halves to perform the exploratory and confirmatory factor analyses, we detected how the sample size slightly influenced the results in terms of fit (the CFA obtained a better fit with the whole sample). This relatively limited sample size also prevented us from establishing a broad scaling in terms of percentiles and T-scores. Therefore, future studies should provide more fine-grained scaling values based on more representative samples. In addition to this, the sex distribution was not homogeneous (85.1% of participants were females). While this may be seen as a limitation, it should be noted that, in the general population, EDs are also more prevalent in women [11]. Moreover, the number of women studying university degrees is higher compared with the number of men (59.8%). Similarly, there is also a higher percentage of women studying Health Sciences degrees, such as Psychology (71.2% of women), which are the degrees from which the participants were mainly recruited [42]. Another limitation is related with the difficulty in determining whether the students met diagnostic criteria for a mental disorder or not. To do this, we asked the following question within the evaluation items “Are you currently receiving any type of psychological treatment?”. It is indeed very challenging to conduct an adequate and complete mental health diagnostic screening during an online survey. However, it would be interesting if future investigations using non-clinical samples administered a telephone interview to ascertain the presence or absence of diagnostic criteria. This was not performed in the present study because this would require obtaining personal data from the participants (name and telephone number), which would have probably reduced their willingness to participate due to anonymity loss. Obtaining a relatively large sample size was preferred in this case. Finally, it should be noted that this research was developed with a very specific sample, that is, university students. Therefore, as noted throughout the text, the results must be interpreted with caution and may not be generalizable to other populations.

## 5. Conclusions

The interest in transdiagnostic/dimensional approaches for the assessment and treatment of EDs have been the basis of the development of the MEDI. This is a 49-item inventory which has been created based on the categorical−dimensional assessment of EDs suggested by Brown and Barlow [6]. Its nine-dimensional structure represents the main clinical features shared by people with EDs diagnosis including two higher order dimensions (negative and positive temperament (NT and PT)) and seven lower order dimensions (depressive symptoms (DM), anxiety symptoms (AA, SOM, SOC, IC and TRM), and avoidance strategies (AVD)) [10]. Our Spanish adaptation study of the MEDI in a non-clinical sample of university students confirmed this nine-dimensional structure of the MEDI and contributed in enhancing the evidence as to its validity and good psychometric properties. This is the first study of the MEDI in a non-American sample, which provides further support for the multicultural properties of this inventory and offers counselors, clinicians, and researchers working with university students in the Spanish language a new inventory to quickly assess the main clinical dimensions of EDs. This comprehensive dimensional assessment of EDs could facilitate the professional decision-making about the type of intervention, prevention, or treatment that is the best option for university students seeking help.

The data provided in this study will be especially useful for clinicians and researchers working from a transdiagnostic perspective. For example, those using the Unified Protocol for transdiagnostic treatment of EDs [43], will be able to compare the MEDI scores obtained before and after the preventive or treatment intervention. This could be used to evaluate the effect of the treatment in each of the nine dimensions of the MEDI and adapt the treatment according to the dimensions in which the patient is responding more poorly. In addition, we also believe that the results will be of interest for health professionals working with clinical samples with EDs, as they will be able to compare the scores obtained by their patients with those obtained in the present study (non-clinical sample) to establish desirable therapeutic goals.

To the best of our knowledge, this is the first study to evaluate the validity evidence of the MEDI since the study in which the instrument was developed. Therefore, we want to encourage researchers to further investigate this validity evidence in samples of different characteristics (non-clinical, subclinical, and clinical), contexts (university settings, community settings, public clinical settings, etc.), formats of delivery (pencil and paper and online) and countries/languages. It will be interesting to see, for example, whether the nine-factor structure is replicated in clinical samples or if a four-factor structure emerges as a potential solution in community samples. Another aspect already mentioned by Rosellini and Brown [6] in their validation is the importance of evaluating the temporal stability of the scale in future studies, since the item scale does not refer to a specific time. In this sense, it would also be interesting to apply the scale before and after having received a psychological intervention in order to assess its sensitivity to therapeutic change.

We encourage researchers to work in this direction, especially given the crisis of categorical diagnostic systems and the urge to shift into dimensional or hybrid approaches.

## Figures and Tables

**Figure 1 ijerph-18-08251-f001:**
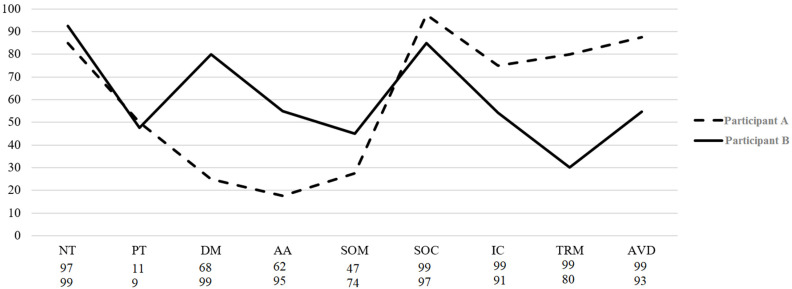
Profiles of two participants according to their scores in each MEDI. Legend. The Y-axis represents the percentage of the score obtained in each dimension over the maximum score. Scores under each dimension represent the participant’s percentile. NT: Neurotic temperament; PT: Positive temperament; DM: Depressed mood; AA: Automatic arousal; SOM: Somatic anxiety; SOC: Social anxiety; IC: Intrusive cognitions; TRM: Traumatic re-experiencing; AVD: avoidance.

**Table 1 ijerph-18-08251-t001:** Sociodemographic characteristics of the participants (*n* = 455).

Sociodemographic Characteristics	*n*	%
Marital status		
Single	293	64.4
Married or in a relationship	156	34.3
Divorced	5	1.1
Widowed	1	0.2
Employment status		
Inactive (retired, unemployed, sick leave)	309	67.9
Active	146	32.1
Current university studies		
Psychology	245	53.8
Master’s in Psychology	58	12.7
Nursing	29	6.4
Doctorate	27	5.9
Teaching children	17	3.7
Medicine	13	2.9
Degree in another discipline	55	12.1
Master’s in another discipline	11	2.4
History of psychological/psychiatric treatment		
No	299	65.7
Yes	156	34.3

**Table 2 ijerph-18-08251-t002:** Instruments to evaluate the validity of MEDI constructs.

	Rosellini and Brown (2019)	This Study	Items
Neurotic temperament	Neuroticism subscale from the NEO-Five Factor Inventory (NEO-FFI; Costa and McCrae, 1992)	Neuroticism subscale from the NEO-Five Factor Inventory (NEO-FFI; Costa and McCrae, 1992); Spanish adaptation by TEA Ediciones (1999)	12
Positive temperament	Extraversion subscale from the NEO-Five Factor Inventory (NEO-FFI; Costa and McCrae, 1992)	Extraversion subscale from the NEO-Five Factor Inventory (NEO-FFI; Costa and McCrae, 1992); Spanish adaptation by TEA Ediciones (1999)	12
Depressed mood	Depression scale from the Depression Anxiety Stress Scales (DASS-21; Lovibond and Lovibond, 1995)	Depression scale from the Depression Anxiety Stress Scales (DASS-21; Lovibond and Lovibond, 1995); Spanish adaptation by Bados et al. (2005)	7
Automatic arousal	Anxiety scale from the Depression Anxiety Stress Scales (DASS-21; Lovibond and Lovibond, 1995)	Anxiety scale from the Depression Anxiety Stress Scales (DASS-21; Lovibond and Lovibond, 1995); Spanish adaptation by Bados et al. (2005)	7
Somatic anxiety	*DSM–5* diagnoses (assessed by the ADIS-5)	The Physical Anxiety scale from the Anxiety Sensitivity Index-3 (ASI-3; Taylor et al., 2007); Spanish adaptation by Sandín et al. (2007)	6
Social anxiety	Social Interaction Anxiety Scale (SIAS total score; Mattick and Clarke, 1998)	Brief version of the Fear of Negative Evaluation Scale (BFNE; Leary, 1983); Spanish adaptation by Gallego-Pitarch (2010)	8
Intrusive cognitions	Obsessing scale of the Revised Obsessive–Compulsive Inventory (OCI-R; Foa et al., 2002)	Obsessing scale of the Revised Obsessive–Compulsive Inventory (OCI-R; Foa et al., 2002); Spanish adaptation by Fullana et al. (2004)	18
Traumatic re-experiencing	*DSM–5* diagnoses (assessed by the ADIS-5)	Davidson Trauma Scale (DTS; Davidson et al., 1996); Spanish adaptation by Bobes et al. (2000)	18
Total subscales items			88

**Table 3 ijerph-18-08251-t003:** Factor loadings and discrimination indices of the MEDI items (*n* = 455).

Items	NT	PT	DM	AA	SOM	SOC	IC	TRM	AVD
1.	0.55 (0.47)								
2.		0.49 (0.41)							
3.			0.77 (0.68)						
4.				0.56 (0.50)					
5.							0.63 (0.61)		
6.					0.56 (0.45)				
7.						0.85 (0.81)			
8.								0.83 (0.74)	
9.									0.51 (0.46)
10.	0.71 (0.65)								
11.			0.89 (0.80)						
12.							0.78 (0.69)		
13.				0.78 (0.68)					
14.						0.74 (0.72)			
15.									0.64 (0.61)
16.	0.72 (0.59)								
17.		0.72 (0.57)							
18.				0.68 (0.55)					
19.					0.36 (0.42)				
20.								0.64 (0.60)	
21.							0.82 (0.75)		
22.						0.80 (0.76)			
23.									0.46 (0.37)
24.		0.87 (0.68)							
25.			0.69 (0.65)						
26.				0.62 (0.55)					
27.									0.74 (0.58)
28.					0.76 (0.61)				
29.								0.84 (0.77)	
30.							0.72 (0.65)		
31.									0.52 (0.49)
32.	0.70 (0.61)								
33.		0.51 (0.48)							
34.									0.43 (0.39)
35.	0.63 (0.57)								
36.		0.45 (0.43)							
37.			0.78 (0.75)						
38.					0.72 (0.70)				
39.								0.79 (0.72)	
40.							0.75 (0.71)		
41.						0.86 (0.81)			
42.									0.53 (0.41)
43.			0.64 (0.58)						
44.				0.67 (0.61)					
45.					0.58 (0.40)				
46.							0.71 (0.68)		
47.						0.94 (0.88)			
48.								0.70 (0.65)	
49.									0.74 (0.43)
α	0.80	0.74	0.87	0.79	0.75	0.92	0.87	0.87	0.77

**Note**. NT: Neurotic temperament; PT: Positive temperament; DM: Depressed mood; AA: Automatic arousal; SOM: Somatic anxiety; SOC: Social anxiety; IC: Intrusive cognitions; TRM: Traumatic re-experiencing; AVD: Avoidance factor loadings (discrimination indices—corrected item−test correlation). α: Cronbach’s alpha loadings under 0.30 are not shown.

**Table 4 ijerph-18-08251-t004:** Sex differences in MEDI scales total scores.

	Total Sample *n* = 455	Females *n* = 387	Males *n* = 66	F	*p*-Value
Neurotic temperament	17.97 (8.86)	18.33 (8.76)	15.86 (9.24)	4.40	0.036
Positive temperament	28.49 (6.51)	28.80 (6.40)	27.03 (6.73)	4.27	0.039
Depressed mood	9.02 (8.46)	8.63 (8.24)	10.98 (9.37)	4.43	0.039
Automatic arousal	7.45 (7.74)	7.55 (7.86)	6.88 (7.09)	0.42	0.518
Somatic anxiety	13.34 (7.68)	13.51 (7.70)	12.41 (7.66)	1.15	0.285
Social anxiety	13.88 (10.22)	14.03 (10.13)	12.58 (10.38)	1.15	0.284
Intrusive cognitions	9.98 (9.99)	9.70 (9.89)	11.42 (10.52)	1.68	0.196
Traumatic re-experiencing	7.11 (7.99)	7.16 (7.93)	6.45 (8.20)	0.45	0.505
Avoidance	19.64 (10.93)	19.82 (10.67)	18.48 (12.52)	0.84	0.360

**Note.** Mean (standard deviation).

**Table 5 ijerph-18-08251-t005:** Normative data for the MEDI scales.

	NT ^1^	PT ^1^	DM ^1^	AA	SOM	SOC	IC	TRM	AVD
PC 25									
Direct scores	11/8	24/24	2/4	1	8	5	2	1	11
T-scores	43/41	49/48	35/40	35	41	38	36	35	35
PC 30									
Direct scores	12/9	26/25	3/4	2	8	6	3	1	13
T-scores	44/42	51/49	36/40	36	41	36	36	35	36
PC 50									
Direct scores	18/15	30/27	6/8	5	12	12	6	4	19
T-scores	49/48	56/51	39/44	39	44	44	39	37	38
PC 60									
Direct scores	21/17	31/28	8/10	7	14	15	9	7	22
T-scores	51/50	57/52	41/46	41	46	46	41	40	39
PC 75									
Direct scores	25/23	34/31	13/15	11	19	22	46	11	27
T-scores	55/55	60/56	46/51	45	50	52	47	44	40
PC 90									
Direct scores	30/30	37/36	21/30	19	23	29	25	19	33
T-scores	59/62	63/62	53/66	53	53	58	54	51	42
PC 99									
Direct scores	36/34	39/38	33/32	30	37	38	40	33	50
T-scores	64/66	65/64	65/68	64	65	66	66	64	47

**Note.** NT: Neurotic temperament; PT: Positive temperament; DM: Depressed mood; AA: Automatic arousal; SOM: Somatic anxiety; SOC: Social anxiety; IC: Intrusive cognitions; TRM: Traumatic re-experiencing; AVD: avoidance. ^1^ Scores are shown for female/male.

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
