# Peer review of "Validity Evidence of the Multidimensional Emotional Disorders Inventory among Non-Clinical Spanish University Students"

_ijerph, 2021, doi:10.3390/ijerph18168251_

Round 1

Reviewer 1 Report

The manuscript Validity Evidence of the Multidimensional Emotional Disorders Inventory among non-clinical Spanish University Students aims to adapt and explore the sources of validity evidence of the MEDI in a sample of Spanish university students. Different types of factor analyses were conducted and indices for validity measures were reported. The authors report a 9-factor structure to be most suitable. Although psychometric properties seem to be convincing, slight changes and some clarification are needed before publication.

1.) The authors stress the need for the development of alternative measures for the categorical assessment of mental health disorders (e.g., dimensional or hybrid dimensional-categorical measures). At the same time, they report that so far, there is only little information available on the validity of such mixed approaches to the assessment of psychopathology. The authors may describe this limited evidence available in more detail rather than just to paraphrase it. Is the original MEDI already in use? Is there evidence available for clinical populations? The original version has been published in 2018 already. Why should it be translated if no one uses it?

2.) It is unclear why the authors refer to two higher and some lower order dimensions for the 9 MEDI dimensions. In the factor analysis literature this would suggest that 7 factors are clustered within two higher order factors. This does not seem to be the case here. After skimming through the original MEDI article, it also seems not to be the case for the original MEDI (doi: 10.1037/pas0000649).

3.) Some remarks concerning the methods/results:

  • 6, 2.4. Data analysis, ll. 227/228: please provide conventional values/cutoff values for the goodness of fit measures used.
  • The authors may want to consider restructuring their results and describe the results in the order they obtained them: e.g., p. 6, ll. 242/243: indices assessing the suitability of the data for data reduction techniques (Kaiser-Meyer-Olkin test, Barlett test) are mentioned along with the goodness of fit indices of the final EFA model and before results of the parallel plot.
  • Table S1: which of the 9 original factors comprise the 4 factor solution? e.g., did specific subscales of the 9 factors fall into specific subscales of the 4 factor solution? What is the relation between the 9-factos and the 4-factors?
  • Was there a reason as to why the authors do not report RMSR values for the CFA but for the EFA? In addition, the Root Mean Square Error of Approximation may be reported as well, as this index attempts to correct for model complexity and sample size.
  • 7, l. 255: What exactly is meant by “modification indices”. Please describe those indices. Why should the error terms of item 19, 38 and item 49 and factor 9 be correlated? What does it mean to correlate the error terms of one entire factor with an item? Was this also done for the original English inventory? Moreover, the authors applied two model modifications in one step (correlating error terms and increasing the sample size) and conclude that increasing the sample size had an effect of the goodness of fit value. The authors may include the steps individually and report CFI values/changes in the CFI after each step.
  • Table 3: names of the subscales may be added. What was the cutoff value for the item loadings? Which index was used as “discriminant indices”?
  • Minor: Figure 1: the legend should have a more intuitive name (e.g., participant A, participant B)

4.) the authors hypothesize to replicate the original 9-factor structure. In fact, they could retain it and even show that this model is superior compared to a 4-factor solution. Why is the 4-factor solution still thoroughly presented? Why may it be used? Also what original factors comprise this model? Is it possible to label these 4 factors? Without labeling it, it’s use may not great in clinical practice.

Author Response

First of all, we would like to thank the reviewers for all their comments, which have notably improved the clarity and quality of the manuscript. Below are the responses to the comments received.

Reviewer 1

The manuscript Validity Evidence of the Multidimensional Emotional Disorders Inventory among non-clinical Spanish University Students aims to adapt and explore the sources of validity evidence of the MEDI in a sample of Spanish university students. Different types of factor analyses were conducted and indices for validity measures were reported. The authors report a 9-factor structure to be most suitable. Although psychometric properties seem to be convincing, slight changes and some clarification are needed before publication.

1.)   The authors stress the need for the development of alternative measures for the categorical assessment of mental health disorders (e.g., dimensional or hybrid dimensional-categorical measures). At the same time, they report that so far, there is only little information available on the validity of such mixed approaches to the assessment of psychopathology. The authors may describe this limited evidence available in more detail rather than just to paraphrase it. Is the original MEDI already in use? Is there evidence available for clinical populations? The original version has been published in 2018 already. Why should it be translated if no one uses it?

Response: Thank you very much for highlighting these concerns. First, regarding to the use of the original MEDI, to our knowledge the MEDI has been used in two scientific articles. One with its complete version (Boettcher et al., 2020) and another using some of its subscales (Castro-Camacho et al., 2021). The article by Boettcher et al. (2020) presents how to use the MEDI in a clinical case, in order to give visibility to the clinical utility of the self-report measure. The study by Castro-Camacho et al. (2021) explores the effects of a preventive intervention based on the Unified Protocol to deal with the transdiagnostic vulnerability factors for the development of emotional disorders, in a sample of university students from the city of Bogotá. For this purpose, the authors use some of the subscales of the MEDI. The only study that has used the MEDI in a clinical sample is the one where the instrument was originally developed and validated, less than two years ago (Rosellini & Brown, 2019). The main explanation for the small number of articles using this questionnaire is firstly, due to its recent publication. The original version of the article was published in 2019, one year before the onset of the pandemic caused by SARS-COVID-2, which may have impacted the development of clinical trials, particularly delaying or paralyzing the recruitment of participants and the subsequent publication of research articles. Despite this circumstance, it is in our knowledge that a research group at the University of Coimbra have translated and conducted a validation study of the MEDI in a community sample in Portugal. We believe that this self-report is going to receive more interest in the following 2 years, especially thanks to validation studies such as the present one, which allow the measure to be used in different countries. In line with this, for the MEDI to be used in clinical practice or in different evaluation contexts, it is necessary to increase the evidence available for this instrument. This necessarily involves the translation and validation of the instrument in different samples from different countries (languages), in order to be implemented. This is why we believe the present study is important for research and clinical practice purposes.

2.)   It is unclear why the authors refer to two higher and some lower order dimensions for the 9 MEDI dimensions. In the factor analysis literature this would suggest that 7 factors are clustered within two higher order factors. This does not seem to be the case here. After skimming through the original MEDI article, it also seems not to be the case for the original MEDI (doi: 10.1037/pas0000649).

Response: Thank you very much for the comment. We agree with the reviewer that the information provided in the introduction section can be confusing because of the differentiation between lower and higher order factors. This was made from the dimensional-categorical hybrid approach proposed by Brown and Barlow (2009), but this is certainly not included in the validation of the Multidimensional Emotional Disorder Inventory (Rosellini & Brown, 2019). To avoid confusion, we will modify the information in paragraph 5 of the introduction, on page 2, including only the factors that are included in the MEDI without differentiating their hierarchical order.

Modified text line 65: Considering the empirical limitations of purely categorical models and the practical limitations of dimensional approximations, Brown and Barlow [10] developed a hybrid dimensional-categorical approach to the assessment of EDs. This was done in an attempt to offer a mixed vision that combined the fundamental advantages of both approach-es. Specifically, Brown and Barlow chose to differentiate a series of factors shared by EDs. They argued that, based on the scores of such factors, they would be able to reveal a unique profile for each patient, which may be closer or more distant to the diagnostic categories provided by traditional classification systems. These factors are neuroticism, anxiety or be-havioral inhibition (NT), behavioral activation or positive affect (PT), depressed mood (DM), autonomic arousal (AA), somatic anxiety (SOM), social anxiety (SEC), intrusive cognitions (IC), traumatic re-experiencing and dissociation (TRM), and avoidance (AVD) [10].

3.) Some remarks concerning the methods/results:

  • 6, 2.4. Data analysis, ll. 227/228: please provide conventional values/cutoff values for the goodness of fit measures used.

Response: Recommended cut-offs values are now provided along with the references supporting them.

  • The authors may want to consider restructuring their results and describe the results in the order they obtained them: e.g., p. 6, ll. 242/243: indices assessing the suitability of the data for data reduction techniques (Kaiser-Meyer-Olkin test, Barlett test) are mentioned along with the goodness of fit indices of the final EFA model and before results of the parallel plot.

Response: As suggested, we have now restructured the results and reported the goodness of fit indices after reporting the suitability of data and the factor solution.

  • Table S1: which of the 9 original factors comprise the 4 factor solution? e.g., did specific subscales of the 9 factors fall into specific subscales of the 4 factor solution? What is the relation between the 9-factors and the 4-factors?

Response: Thank you very much for this input. No details about the 4-factor structure were given because the statistical analyzes showed a better fit of the original structure. However, it is true that this information can be interesting, so we have added it in the supplementary material, just behind Table S1. The text appears on page 13.

Added text line 426: Table S1 shows the items that belong to each of the 4 factors of the structure proposed by the exploratory factor analysis. The relationship between the 4-factor structure and the original 9-factor structure [6] is shown below. Factor 1 is made up of 7 items from the original avoidance subscale (AVD), 5 items from the traumatic re-experiencing scale (TRM), 4 items from the intrusive cognitions scale (IC), and 1 item from autonomic arousal scale (AA). Factor 2 is made up of 5 items from the original social anxiety subscale scale (SOC) and 2 items from the avoidance scale (AVD). Factor 3 is composed of 5 items from the positive temperament scale (PT), which have a negative loading, and 5 items from the depressed mood scale (DM). Finally, Factor 4 is made up of 5 items from the neurotic temperament (NT) subscale, 4 items from the autonomic arousal scale (AA), 2 items from the intrusive cognitions scale (IC), and 5 items from the so-matic anxiety scale (SOM).

  • Was there a reason as to why the authors do not report RMSR values for the CFA but for the EFA? In addition, the Root Mean Square Error of Approximation may be reported as well, as this index attempts to correct for model complexity and sample size.

Response: We apologize for the missing information. We now report both the RMSR and RMSEA indices in both CFA analysis.

  • 7, l. 255: What exactly is meant by “modification indices”. Please describe those indices. Why should the error terms of item 19, 38 and item 49 and factor 9 be correlated? What does it mean to correlate the error terms of one entire factor with an item? Was this also done for the original English inventory? Moreover, the authors applied two model modifications in one step (correlating error terms and increasing the sample size) and conclude that increasing the sample size had an effect of the goodness of fit value. The authors may include the steps individually and report CFI values/changes in the CFI after each step.

Response: We have now explained what the modification indices are, as well as why controlling for them is an adequate procedure to improve CFA models (see methods section). We now also report CFI values and changes after using the whole sample and after including the modification indices.

  • Table 3: names of the subscales may be added. What was the cutoff value for the item loadings? Which index was used as “discriminant indices”?

Response: We have added the required information

  • Minor: Figure 1: the legend should have a more intuitive name (e.g., participant A, participant B)

Response: We totally agree with the reviewer. We have added information following this suggestion.

4)   the authors hypothesize to replicate the original 9-factor structure. In fact, they could retain it and even show that this model is superior compared to a 4-factor solution. Why is the 4-factor solution still thoroughly presented? Why may it be used? Also what original factors comprise this model? Is it possible to label these 4 factors? Without labeling it, it’s use may not great in clinical practice.

Response: As the reviewer points, the original structure showed better fit than the alternative 4-factor solution. However, we decided to report both structures as well as the comparison between fit indices for several reasons: First, there is still few available evidences of the structural validity of the original 9-factor solution. Second, the nature of the sample is different from the original study and the international quality standard recommends to perform exploratory analysis before cross-validating the structure (International Test Commision, 2017; Izquierdo et al., 2014). Finally, the goodness of fit criteria of the alternative 4-factor solution yielded adequate values, suggesting its suitability. Therefore, since very few evidence still exists on the MEDI’s factor structure, we opted to report the alternative 4-factor solution in the supplementary material. Its report as supplementary material aims at acknowledging the potential alternative structures of the MEDI, but also aim to adhere to the recommendation of the 5th guideline for psychological assessment and evaluation of the American Psychological Association (APA, 2020).

Reviewer 2 Report

The manuscript deals with an interesting topic, i.e. the validation of the Multidimensional Emotional Disorders Inventory (MEDI) in Spanish University students. The study is interesting and was conducted with a sound methodology. The contribution is relevant not only to Spanish scholars, but to all those involved in the debate between categorical and dimensional approaches to diagnosis in psychiatry.

Globally, the paper is well written and an interesting read.

I have only one major point to raise. In the Methods, the authors state that "The research was approved by the Research and Ethics Committee of the (blinded) University". On the other hand, at the end of the paper the authors state that "Institutional Review Board Statement: Not applicable". This is not clear.

Besides that, I note that in the abstract MEDI is miss-spelled MIDI (line 20). 

Moreover, the authors should discuss the fact that the vast majority of involved subjects were females or at least mention this as a limitation of the study (a paragraph on study limitations is needed).

Author Response

First of all, we would like to thank the reviewers for all their comments, which have notably improved the clarity and quality of the manuscript. Below are the responses to the comments received.

Reviewer 2

 The manuscript deals with an interesting topic, i.e. the validation of the Multidimensional Emotional Disorders Inventory (MEDI) in Spanish University students. The study is interesting and was conducted with a sound methodology. The contribution is relevant not only to Spanish scholars, but to all those involved in the debate between categorical and dimensional approaches to diagnosis in psychiatry.

Globally, the paper is well written and an interesting read.

I have only one major point to raise. In the Methods, the authors state that "The research was approved by the Research and Ethics Committee of the (blinded) University". On the other hand, at the end of the paper the authors state that "Institutional Review Board Statement: Not applicable". This is not clear.

Response: Thank you for the kind words on our work. This was a misunderstanding which is now corrected:

Modified text line 509: Institutional Review Board Statement: The study was conducted according to the guidelines of the Declaration of Helsinki, and approved by the Ethics Committee of blind note.

Besides that, I note that in the abstract MEDI is miss-spelled MIDI (line 20).

Response: Thank you very much for highlighting this mistake. We have corrected it.

Moreover, the authors should discuss the fact that the vast majority of involved subjects were females or at least mention this as a limitation of the study (a paragraph on study limitations is needed).

Response: Thank you very much for the comment. We fully agree with the reviewer. We have added the following paragraph in the limitations section:

Added text line 409: In addition to this, the sex distribution was not homogeneous (85.1% of participants were females). While this may be seen as a limitation, it should be noted that, in the general population, EDs are also more prevalent in women [11]. Moreover, the number of women studying university degrees is higher compared with men (59.8%). Similarly, there is also a higher percentage of women studying Health Sciences degrees like Psyhology (71.2% of women), which are the degrees where the participants were mainly recruited [43].